# The Phenotypic Responses of Vascular Smooth Muscle Cells Exposed to Mechanical Cues

**DOI:** 10.3390/cells10092209

**Published:** 2021-08-26

**Authors:** Lise Filt Jensen, Jacob Fog Bentzon, Julian Albarrán-Juárez

**Affiliations:** 1Atherosclerosis Research Unit, Department of Clinical Medicine, Aarhus University, 8200 Aarhus, Denmark; lfj@clin.au.dk (L.F.J.); jfbentzon@clin.au.dk (J.F.B.); 2Experimental Pathology of Atherosclerosis Laboratory, Spanish National Center for Cardiovascular Research (CNIC), 28029 Madrid, Spain; 3Steno Diabetes Center Aarhus, Department of Clinical Medicine, Aarhus University, 8200 Aarhus, Denmark

**Keywords:** smooth muscle cells, mechanical forces, cyclic stretch, stiffness, extracellular matrix, phenotypic modulation

## Abstract

During the development of atherosclerosis and other vascular diseases, vascular smooth muscle cells (SMCs) located in the intima and media of blood vessels shift from a contractile state towards other phenotypes that differ substantially from differentiated SMCs. In addition, these cells acquire new functions, such as the production of alternative extracellular matrix (ECM) proteins and signal molecules. A similar shift in cell phenotype is observed when SMCs are removed from their native environment and placed in a culture, presumably due to the absence of the physiological signals that maintain and regulate the SMC phenotype in the vasculature. The far majority of studies describing SMC functions have been performed under standard culture conditions in which cells adhere to a rigid and static plastic plate. While these studies have contributed to discovering key molecular pathways regulating SMCs, they have a significant limitation: the ECM microenvironment and the mechanical forces transmitted through the matrix to SMCs are generally not considered. Here, we review and discuss the recent literature on how the mechanical forces and derived biochemical signals have been shown to modulate the vascular SMC phenotype and provide new perspectives about their importance.

## 1. Introduction

### Mechanical Forces and Smooth Muscle Cells

Cardiovascular diseases are one of the leading causes of global death in developing countries [1]. More than 80% of cardiovascular disease-associated mortality is attributable to atherosclerosis, a chronic inflammatory disease of the vessel wall [2]. During the development of atherosclerosis and other cardiovascular diseases, vascular smooth muscle cells (SMCs) continuously shift from a contractile state towards other phenotypes that differ substantially from differentiated SMCs. This process is characterized by a reduced or lost expression of contractility-associated proteins, increased expression of marker genes associated with other cell types, including matrix-remodeling enzymes, and increased production of alternative types of extracellular matrix (ECM) proteins [3]. As a result, the modulated SMCs typically undergo a burst of proliferation and migration, leading to the development of clonal SMC populations in expanding lesions [4]. Efforts have been made to identify the environmental cues, signaling pathways, and mechanisms that maintain the vascular SMCs in their contractile phenotypes and how these are disrupted during disease states, but the processes are still far from being understood.

During life, cells in the vasculature are continuously exposed to different mechanical forces that regulate their function and homeostasis. These forces include fluid shear stress, cyclic stretch, and hydrostatic pressure [5]. Fluid shear stress is the frictional force from the blood flow, along with the vascular endothelial layer [6,7]. Endothelial cells lining the vessels can sense changes in the blood flow, convert them into biochemical signals, and trigger cellular responses [8]. The importance of mechanical forces on endothelial function has been extensively reviewed [9,10]. Disturbed shear stress influences the site selectivity of atherosclerotic plaque formation and vessel wall remodeling [11]. In healthy vasculature, the endothelium layer prevents the exposure of SMCs located in the vascular media to shear stress. The pulsating nature of the blood flow driven by the heart generates a cyclic stretch, which acts on the medial layers of the vasculature rich in SMCs but can also modulate the endothelial function [12,13]. Early studies in the field have defined that the human aorta experiences approximately a 10% cyclic stretch elongation on its external diameter within each heartbeat under physiological conditions [14,15]. However, during pathological conditions, including atherosclerosis and acute hypertension, blood vessels experience high-magnitude stretches of 20% and above [13,16]. In addition to the shear stress and cyclic stretch, all cellular layers in the vascular wall are subjected to compressive forces in vivo. However, the mechanisms by which compressive forces influence vascular cell phenotypes have not been fully elucidated [17]. Thus, mechanical forces are critical to maintaining a normal and healthy vasculature. However, the loss or excess of these mechanical cues can be detrimental and predisposed to vascular diseases.

Given the critical role of SMC phenotypic modulation in the pathogenesis of vascular diseases, understanding how SMCs sense and transduce mechanical forces into biochemical signals and their interactions with surrounding ECM proteins is critical to prevent the development and progression of vascular diseases. However, the importance of mechanical forces for controlling the cellular phenotype in vascular cells has largely been neglected in previous in vitro research, particularly on SMCs. Furthermore, unlike the native vessel, the substrate (a rigid and static plastic culture plate) used in most studies of in vitro culture lacks a physiological microenvironment. Therefore, in this review, we discuss recent 2D in vitro studies that evaluated the effect of mechanical forces (cyclic stretch) on vascular SMCs. We particularly spotlight the effects of these forces on their interaction with ECM proteins, the expression of smooth muscle (SM) contractile and synthetic marker genes, and other key aspects of SMC function, such as migration and proliferation.

## 2. How Do We Measure Smooth Muscle Phenotypic Modulation?

It is widely accepted that, when SMCs are removed from their native environment and placed in a culture, they undergo a similar shift in cell phenotype as observed in several vascular diseases [18]. This phenotypic modulation is a complex and graded process [19]. The changes that SMCs undergo range from subtle variations in contractility, where the cells still express contractile proteins, to more profound changes in SMC morphology, function, and gene expression patterns. Some SMCs lose the SMC phenotype altogether, and the cells are transformed to resemble other mesenchymal cell types, such as fibroblast-like or osteochondrogenic-like cells [20].

In vitro, the loss of the SMC contractile phenotype has, in most studies, been measured as reduced mRNA or protein levels of classical contractile marker genes, such as actin alpha 2 smooth muscle (*ACTA2*), calponin (*CNN1)*, smoothelin (*SMTN)*, smooth muscle myosin heavy chain (*MYH11)*, transgelin (*TAGLN)*, etc. A reduction in the expression of SMC contractile markers is often accompanied by increased cell migration and proliferation [21]. Most of the in vitro studies investigating the link between mechanical forces and SMC phenotypic modulation focused on evaluating the gene expression, migration, or cell proliferation. However, only a few studies evaluated all at the same time. Therefore, the conclusions about a specific SMC phenotypic state based solely on one of the classical signals, such as a decreased expression of a few SM contractile marker genes, without being complemented with other readouts of cellular functions, such as contractility, migration, and proliferation, must be taken cautiously.

## 3. Vascular Mechanical Microenvironment

### 3.1. Smooth Muscle Phenotype and Extracellular Matrix Proteins

A significant function of SMCs in the arterial media of large vessels is to synthesize and organize a unique and highly elastic ECM to deal with the mechanical forces imposed by the pulsatile blood flow [22]. The ECM can be defined as the noncellular component that surrounds vascular cells and other organs. It is mainly formed from filamentous and sheet-forming proteins, proteoglycans, and glycosaminoglycans [23]. Recent advances in genetics and proteomics have facilitated our understanding of ECM proteins as potential novel disease biomarkers [24]. SMCs are surrounded by a basement membrane of ECM proteins predominantly formed by laminins, collagens IV and XVIII, perlecan, and other ECM components in healthy vessels. SMCs are also embedded in fibrillar collagen types I, III, and V; fibronectin; and other proteoglycans within the arterial media, known as transitional or interstitial ECM [23]. The changes in ECM degradation and production are recognized as hallmarks of vascular diseases and likely guide the loss of the SMC phenotype and modulation of alternative phenotypes [25]. For example, an altered vascular wall triggers a series of events characterized by the excessive production of poorly organized and highly stiffer collagens (such as collagen I) and other ECM components, such as fibronectin, biglycan, and lumican. These changes in the ECM microenvironment predispose the phenotypic modulation of SMCs [22,25,26]. In vitro work has shown that rat aortic SMCs cultured on collagen I or fibronectin substrates can switch from a contractile to a synthetic phenotype [26,27].

In contrast, rat SMCs cultured on laminin or collagen IV, central components of the basement membrane, maintained a more contractile phenotype, at least for some time [26,27]. Similar observations were made with rabbit [28] and human aortic SMCs [29]. Hence, the SMC phenotype appears to be partly instructed by interactions with the surrounding ECM proteins (Figure 1A).

### 3.2. Influence of Stiffness on Smooth Muscle Phenotype

Changes in the composition of ECM proteins such as elastin and collagen fibers trigger the development of a process known as vascular stiffening. Vascular stiffening increases with age and pathological states such as hypertension and atherosclerosis, as it is often accompanied by fibrosis and increased calcification [30]. To evaluate the effects of stiffness on cellular functions, most in vitro studies culture cells on tunable (soft and hard) gels that can mimic the elasticity of most physiological or pathological conditions within the vessels [31]. To determine the stiffness of solid materials, scientists commonly use the Young’s modulus (named after the British scientist Thomas Young). It quantifies the material’s resistance to elastic deformation elicited by a given tensile force. The effect of the force is dependent on the area; therefore, equations are defined in terms of stress (most commonly, Pascals; Pa) and divided by strain and the change in length of the materials (unitless) [32]. For example, in vivo stiffness of a healthy porcine aorta has been reported at 8 kPa, while mouse aorta has been at 5 kPa. On the other hand, atherosclerotic changes in the arteries of ApoE-KO mice elicit an increased stiffness of about 28 kPa [33].

An example of how cells can respond to different stiffness conditions was shown by experiments with human SMCs cultured on soft (2 kPa) and stiff (20 kPa) surfaces for 24 h. The expression of SM contractile marker genes such as *MYH11*, *TAGLN*, *CNN1* and *SMTN* was downregulated on stiff substrates. In addition, the genes associated with the proliferation and the synthetic phenotype of SMC were upregulated on stiff substrates compared to soft ones [34]. In contrast, the transcriptome sequencing analysis of mouse SMCs cultured on soft and stiff gels showed the opposite. SMCs cultured on soft substrates (0.17 kPa) increased the expression of a number of genes involved in the synthetic phenotype, such as osteopontin (*OPN*), vimentin, matrix metalloproteinases, and inflammatory cytokines, in comparison to stiff (1.2 kPa) substrates [35]. Interestingly, a more recent study cultured human aortic SMC in soft (1 kPa), medium (40 kPa), and hard (100 kPa) substrates [36]. They observed that SMC cultured on both soft and stiff substrates increased their expression of macrophage *CD68,* galectin 3 *(LGALS3*), and inflammatory interleukin 6 (*IL-6)* and interleukin 1 beta *(IL1**β**)* markers compared to cells on medium stiffness substrates [36]. Notably, *MYH11* expression, contrary to previous findings, was found upregulated on hard, compared to soft, substrates, thus suggesting that moderate stiffness, a condition closer to the physiological parameters, could be beneficial to SMC function.

Interestingly, the effects on the SMC phenotype elicited by the combination of distinct cues such as different stiffnesses and changes in the ECM proteins associated with stiffening have not been systematically evaluated. Most of the studies have only used gels coated with collagen I or fibronectin to mimic the in vivo microenvironment that SMCs experience in arteries with increased stiffness. For example, a recent study showed that the ECM protein used to coat the gels can differentially affect the SMC phenotype [37]. In this study, the authors observed that rat aortic SMC migration was decreased on stiff gels (103 kPa) coated with collagen I, while it was increased on gels coated with fibronectin [37]. The modulation of the SMC phenotype depends not only on the composition of the ECM but, also, on the physical structure of the matrix presented to the cells. For example, rat aortic SMCs respond with different phenotypes to fibrillar collagen I compared to nonfibrillar collagen I, even though the cell–matrix binding appears to be through the β1 integrin in both cases. It appears that, when collagen fibrils become aligned, the resting tension increases, thus producing a higher Young‘s modulus. As a result, the cells spread more and proliferate faster on stiffer than on flexible fibrils [38]. Efforts have been made to characterize the stiffness-sensitive transcriptome of human SMCs. Bulk RNA sequencing (RNA-seq) of human SMCs cultured on fibronectin-coated soft physiological (4 kPa) or stiff pathological (25 kPa) substrates was performed [39]. While this study identified 3098 stiffness-sensitive genes, they were focused on long non-coding RNAs (lncRNAs) and provided the first transcriptomic landscape of human SMCs in response to stiffness.

As mentioned above, there are important discrepancies in the results of studies examining the influence of substrate stiffness on the SMC phenotype (Figure 1B). It is particularly remarkable that, within the various studies performed, the definition of what is soft and stiff relative to the vascular system is still not completely understood. In addition, how well 2D gels with different stiffness and ECM compositions reflect the in vivo conditions found on standard and stiff arteries remains unanswered. Since the current modeling of stiffness in vitro lacks the external forces found in pulsating arteries, new systems that combine the stiffness and mechanical cues are also needed to identify how significant these mechanical changes are in relation to the SMC phenotype.

## 4. Cyclic Mechanical Stretch

Under normal physiological conditions, SMCs within the vascular wall are primarily exposed to mechanical forces due to the pulsatile nature of blood pressure. To counteract these forces, the vascular tissue stretches circumferentially and longitudinally [12]. SMCs and the surrounding ECM sense and transduce the mechanical stretch into biochemical signals that ultimately control their function. When these mechanical cues are altered, for example, due to the development of arterial diseases, the structural and functional properties of SMCs are modified. Studies have shown that, under physiological conditions, the human aorta undergoes a 10% oscillation on its external diameter [15].

However, when the blood pressure is increased, the degree of stretch also increases, at least until chronic remodeling of the arterial wall changes the average diameter or increases the stiffness of the vascular tissue [12]. In practical terms, a cyclical stretch with amplitudes of 10% and 20% are often applied in cell culture studies to mimic the physiological and high pressure-induced stretch effects, respectively.

### 4.1. In Vitro Modeling of the Cyclic Stretch

Researchers have created diverse in vitro technologies during the last years that, by applying different stretches to SMCs, can mimic in vivo wall distension. Perhaps one of the most widely used systems is the Flexcell^®^ tension system operated through a vacuum pump and regulated by computer software. A uniaxial or equibiaxial strain can be applied to cell monolayers cultured on soft-bottomed silicone elastomer membranes. In contrast to SMCs within blood vessels, cells in a culture are randomly oriented. Since the cellular biochemical responses to uniaxial stretching depend on the cell‘s orientation relative to the direction of stretching [40], the in vitro application of a uniaxial strain may not necessarily reproduce the physiological conditions found in vivo. On the other hand, equibiaxial stretch systems have shown a robust strain homogeneity even on randomly oriented cells in a culture, especially in the central region of the membrane [41]. Other technologies such as Strex or ShellPa are also available, though less used. In these systems, the stretch can be controlled by an electric motor regulated by a control unit. The relatively narrow stretch chambers used by the Strex and ShellPa systems have been proposed to also exert a compressive strain on the cells, which seems to be one disadvantage of these systems compared to the Flexcell system [42].

### 4.2. Effect of the Cyclic Stretch on SMC Marker Gene Expression

Multiple previous studies have evaluated the effects of cyclic stretch on SMC modulation using gene expression as the readout (Table 1 and Figure 2). 

In most of these, SMCs were subjected to stretch using flexible-bottomed plates coated with collagen I. In interpreting these studies, it should therefore be noted that collagen I itself can contribute to the dedifferentiation of SMCs [28], and these culture conditions may therefore not be optimal in mimicking the arterial SMCs subjected to stretch in vivo. Other ECM proteins, such as fibronectin, have also been used—however, less frequently [48,53]. Interestingly, the combination of stretch and culturing on basement ECM substrates, which may potentially support the differentiated SMC phenotype better, have been sparsely explored.

Surprisingly, very few studies have examined the effects of the physiological levels of stretch on human SMCs [43,44,45,46,47]. Two or five days of physiological cyclic stretch (7%, 1 Hz) did not induce significant changes in the SM marker expression levels in human umbilical artery SMCs on collagen I membranes [43].

Other studies have looked at high stretch stimulations (>10%) of human SMCs, which have generally been shown to induce a dedifferentiated phenotype. For instance, human aortic SMCs cultured on collagen I coated-flexible membranes showed a decreased expression of the SM contractile markers *CNN1*, *TAGLN*, *ACTA2*, and myocardin (*MYOCD)* [51]. The Krüppel-like factor 4 (*KLF4),* another marker gene associated with the synthetic phenotype of SMCs, was also upregulated [51]. Similarly, human umbilical artery SMCs cultured on a thick basement membrane matrix (Matrigel) and exposed to 13% of the stretch for 24 h resulted in a substantial reduction of the ACTA2, MYH11, and CNN1 protein levels compared to the static controls [49]. Reduced contractile marker levels were accompanied by the upregulation of some inflammatory genes, interleukin 8 *(IL-8), IL-6, IL1**β*, vascular cell adhesion protein 1 *(VCAM-1),* and intercellular cell adhesion 1 *(ICAM-1*) [49]. The current in vitro studies that subjected human SMCs to stretch differed not only in the mechanical intensity but also in the duration of the stimulation [48,52]. Therefore, new studies exploring these variations on human SMCs are highly needed.

The vast majority of stretch studies found in the literature have used primary aortic rat SMCs (selected examples are shown in Table 1). Some studies have reported the upregulation of contractile markers after rat SMCs were exposed to 10% stretch for 24 h [44,45,46]. These studies suggest that a ≤ 10% stretch intensity (physiological) induced a differentiated state in SMCs even when they were cultured on collagen I or gelatin substrates. However, other work is inconsistent with that idea. When rat aortic SMCs cultured on collagen I were subjected to a 10% stretch (1 Hz) for 24 h, a decreased expression of *Cnn1 and Smtn* and increased levels of the synthetic marker *Opn* were observed [47]. Potential variations between the different studies are the arteries from where the SMC were isolated, the frequency of stretch (1.25 Hz vs. 1 Hz), and the rat strain used (Sprague–Dawley vs. Wistar).

Consistent with the studies of human SMCs, a decrease in the contractile markers *Cnn1*, *Acta2*, and *Tagln* were noted in rat thoracic aorta SMCs subjected to a 15% stretch for 24 h compared to the cells subjected to a 5% stretch [50]. In addition, a study reported no changes in the protein levels of MYOCD when rat aortic SMCs were stimulated with 10% stretch for 30 h. However, *Myocd* expression was upregulated shortly after 6 h when the cells received 20% cyclic stretch. Unfortunately, the ECM protein used to coat the plates in this study was not stated, but this result confirms that a high stretch intensity modulates the gene expression in SMCs [54].

Although the findings are far from consistent, the picture that emerges is that of lower stretch levels retaining SMCs in their contractile state, whereas higher stretch levels (especially >15%) can induce SM phenotypic modulation.

Given the importance of ECM proteins for the behavior of SMCs, there is a knowledge gap regarding the effect of the stretch on cells cultured on other ECM substrates, which are thought to retain the SMCs in the contractile phenotype, such as laminin and collagen IV or others (recombinant engineered proteins) [26,53]. More studies are needed to fully understand the mechanisms behind combining different stretch intensities and various ECM proteins to their effect on SMC phenotypic modulation.

Currently, there is a limited understanding of how the different local distension patterns that occur through different vascular beds and during disease conditions affect the SMC functions. Lee and colleagues have suggested that the type of waveform can modulate the human SMC phenotypic state. For instance, a 24-h cyclic stretch with an amplitude of 7.5% and a waveform that mimics the pressure generated by the heart can induce the upregulation of *CNN1* and *TAGLN* compared to cells stretched with the simple and widely used sinusoidal waveform [55]. It would be valuable to also investigate the effect of the waveforms found in diseased vasculature on the phenotypic modulation of cultured SMCs in these in vitro preparations.

Other in vitro studies that applied a stretch on SMCs have focused on exploring inflammatory genes as a surrogate for dedifferentiation markers. Primary mouse aorta SMCs stimulated by a 15% stretch for 3 h on Flexcell plates coated with collagen I showed increased expression levels of *IL-6* [56]. Another study cultured primary rat aortic SMCs on collagen I membranes (Strex^®^), which were subsequently exposed to a 15% stretch for 4 h or left static [57]. The gene expression changes were analyzed by a microarray. The analysis was mainly focused on potential associations with mitogen-activated protein (MAP) kinase pathways. However, the study showed an upregulation of the genes involved in inflammation, such as the chemokine (C-X-C motif) ligand 1 (*Cxcl1*) and (C-X3-C motif) ligand 1 (*Cx3cl1*) on the cells exposed to a stretch compared to the cells under static conditions.

Similarly, another study performed a microarray analysis of human aortic SMCs after being exposed to a 20% supraphysiological stretch for 24 h compared to cells on static cultures [52]. Their study primarily focused on identifying lncRNA expressions induced by non-physiological stretching. Besides the downregulation of some SM contractile markers, they identified several regulated lncRNAs associated with the tumor necrosis factor and inflammatory signaling pathways. Altogether, these findings provide a link between non-physiological mechanical forces and the inflammatory response elicited by mouse, rat, and human SMCs and potentially highlight the molecular mechanisms of biomechanical stress that contribute to the initiation of atherosclerosis.

### 4.3. Other Aspects of SMC Phenotypic Modulation

In normal, healthy vessels, the majority of SMCs are found in a differentiated or contractile state, are quiescent, and do not migrate. However, during the development of vascular diseases, increased migration and proliferation are important signs of a SMC phenotypic switch that often accompanies the decreased expression of contractile SM marker genes.

#### 4.3.1. Effect of Cyclic Stretch on SMC Migration

The effects of a physiological or supraphysiological stretch on SMC migration has been examined in several studies, but the results have not been consistent (Table 2 and Figure 2).

One recent study investigated the migratory capacity of human aortic SMCs exposed to stretching by the scratch wound-healing assay. The scratch was performed on confluent monolayers before the cells were either subjected to stretching (10%, 1 Hz) or left in static conditions, and the gap closure area was measured at 12 h after the static or stretch stimulation [58]. They reported that the migration of human SMCs cultured on collagen I-coated membranes was reduced compared to the static controls [58]. In contrast, rat SMCs cultured on collagen I-coated membranes were scratched and then left static or subjected to a physiological stretch (10%, 1 Hz for 24 h). Here, the authors observed that a physiological stretch increased the migration capacity of rat SMCs in comparison to static conditions [47]. Taken together, the discrepancy in the results between these studies may be due to minor differences in the type and source of SMC used, the stretch conditions (duration and waveform), and the way the migration capacity was evaluated (during or time after the stretch).

The effects of a supraphysiological stretch have been examined in primary mouse aortic SMCs seeded on collagen I-coated membranes and stimulated with a supraphysiological stretch (20%, 1 Hz). After 3 h, the stretch was stopped, and the scratch was performed. An increased cell migration was seen in stretched compared with static controls [60]. It is possible that such effects are mediated partly by the release of stretch-dependent migratory mediators into the culture media. Indeed, an independent study used the conditioned media of rat thoracic aortic SMCs seeded on Matrigel membranes and exposed to stretching for 24 h (20%, 1 Hz) [54]. They observed that the migratory capacity of the cells that received the conditioned media was increased compared to the control cells fed with the standard media [54]. Further studies are needed to determine the migration capacity of SMCs after long stimulation stretch periods (days) and on other types of ECM substrates.

#### 4.3.2. Effect of Cyclic Stretch on SMC Proliferation

The effect of physiological stretching (< 10%) has been shown to reduce the proliferation of human and rat aortic SMCs compared to the static controls in different studies [58,61,62]. Human SMCs require at least 12 h of physiological stretching to reduce their proliferation [58]. Interestingly, rat aortic SMCs seem to require physiological stretching for more extended periods (48 h to 4 days) to reduce their proliferation [61,62]. However, another study reported an increased proliferation when mouse SMCs were subjected to a 10% physiological stretch for 1 h [63]. One difference in this study was that the SMCs were cultured on gelatin-coated membranes. Therefore, the potential differences of these studies compared to others are the type of ECM protein used, the time of stretch exposure, and SMC source used.

On the other hand, a non-physiological stretch stimulation (> 10%) of either human or rat aortic SMCs has been reported to increase cell proliferation compared to static conditions. For instance, human aortic SMCs that were stretched (> 10%) on collagen I-coated membranes for 12 or 24 h showed increased proliferation compared to cells under static conditions [52,64,65,66,67]. Similarly, proliferation was increased in human umbilical artery SMCs exposed to 13% of stretching for 24 h [49].

Overall, the combined results indicate that a physiological stretching force is quiescent in SMCs, while high-intensity or pathological stretching induces SMC proliferation (Table 3 and Figure 2).

Further studies are required to understand the potential interactions between different types of stretching, including waveforms, and different ECM proteins on SMC proliferation.

#### 4.3.3. Effect of Cyclic Stretch on SMC Apoptosis

Apoptosis or programmed cell death has been identified as an essential process contributing to cardiovascular disease development [68]. It has been found that SMC apoptosis promotes neointimal formation in mice, in part by increasing the cell proliferation, migration, and cell matrix formation [69]. In vitro, various stimuli, including oxidized lipoproteins, alter mechanical stress, and free radicals can induce SMC apoptosis [70]. Studies that exposed mouse SMCs cultured on gelatin to physiological stretching (< 10%) for less than 24 h have reported increased apoptosis compared to static controls [71,72,73]. The increased apoptosis seen in these studies was accompanied by increased proliferation.

The exposure of human aortic SMCs to high-intensity stretching >15% for 12 h showed similar results to those exposed to low-stretch levels. The apoptosis and proliferation were increased compared to the static cultures [65,66]. Additional studies that used high-intensity stretching in human, rat, and mouse SMCs for a range from 4 to 36 h found an increase in apoptosis [74,75,76,77]. A common feature for these studies is that the SMCs were cultured on plates coated with collagen I (Table 4).

In general, studies with human, pig, rat, or mouse SMCs concluded that mechanical stretching increases the level of apoptosis independent of the stretch intensity, duration, ECM coating, and origin of the SMCs [16].

Different laboratories have investigated the mechanisms by which high intensity stretching could induce apoptosis. Some results indicate that the p53 upregulated modulator of the apoptosis protein (PUMA) is induced by interferon-gamma (IFN-γ), the c-Jun N-terminal kinase (JNK), and the interferon regulatory pathway 1 (IRF-1) pathways in response to stretching [76]. A recent study performed a microarray analysis of rat thoracic aorta SMCs exposed to high intensity stretching compared to the static controls using the STREX system [75]. In this study, the authors identified 91 differentially expressed genes, of which 29 were related to cell death. Furthermore, they suggested that inducible nitric oxide synthase (iNOS) expression in rat SMCs protects them from stretch-dependent cell death [75].

## 5. Fluid Shear Stress and SMCs

During vascular diseases or surgical interventions such as angioplasty or endarterectomy, vascular endothelium damage can occur, directly exposing SMCs to different patterns and intensities of shear stress [16]. An early work by the Tarbell lab has shown that, even in an intact artery, at least in some conditions, SMCs are continuously exposed to different shear stress magnitudes due to the interstitial flow driven by the transmural pressure gradient [78]. In vitro studies have demonstrated that SMCs directly react to fluid shear stress [79,80]. Therefore, a deeper understanding of the mechanisms by which fluid shear stress modulates the SMC phenotype represents an important scientific question.

### 5.1. In Vitro Modeling of Fluid Shear Stress

Studies investigating the effects of shear stress have mainly been performed on cultured monolayers of ECs seeded on flat and stiff substrates [9]. The same principles and devices have been applied to studying the effects of shear stress on the SMC phenotype. The most common method is parallel plate flow chambers where the cells are subjected to a constant fluid shear, typically from a warm cell medium moved by a peristaltic pump at a certain speed and pattern. The Ibidi Pump System can mimic various in vivo shear stress conditions, such as the laminar flow typical of atherosclerosis-protected vessels or the oscillatory flow typical of atherosclerosis-prone areas. Parallel plates are made of plastic or glass and coated with various proteins, such as collagens I and IV, and fibronectin. Further studies are required to determine the effect of soft stiffness substrates and different protein substrates on the SMC phenotype in the presence of shear stress.

### 5.2. Shear Stress and Phenotypic Modulation of SMCs

Early studies have used DNA microarrays to determine the global expression profile of human aortic SMCs under fluid shear stress [81]. Cells cultured on glass slides coated with fibronectin were exposed to laminar shear stress (12 dynes/cm^2^) for 24 h and compared to the cells under static conditions. Among the top regulated were the genes involved in the cell cycle and death, cell adhesion, and ECM. In the same study, they confirmed by BrdU labeling that laminar shear stress promotes human SMC proliferation compared to static controls [81]. Unfortunately, no information about the expression of SM contractile marker genes was found or stated in this screen. Multiple other studies, however, have shown that the exposure of rat aortic SMCs to laminar shear stress (8 or 14 dynes/cm^2^) for extended periods of time (15–24 h) reduced the expression of some classical SM markers when compared to static controls, as summarized in Table 4 [80,81,82,83]. In one of these, the exposure of rat cerebral artery SMCs to a laminar flow (15 dynes/cm^2^) for 6, 12, and 24 h resulted in the time-dependent downregulation of *Acta2* and *Tagln*, while matrix metalloproteinase 2 (*Mmp-2)* and tumor necrosis factor-alpha (*Tnf-**α**)* were upregulated. Phenotypic switching in this study was also accompanied by the enhanced proliferation and migration of SMCs after shear stress [83]. Thus, this and other studies suggest that laminar shear stress induces the dedifferentiation of SMCs compared to cells under static conditions. 

However, not all observations fit this concept. One study found that proliferation was decreased rather than increased in rat aortic SMCs exposed to laminar shear stress (14 dynes/cm^2^) for 24 h compared to the static controls [84]. Unfortunately, information about the ECM substrate used as a coating in this study was not stated. Similarly, the exposure of rat aortic SMCs to laminar shear stress (12 dynes/cm^2^) for 24 h decreased the cell proliferation and migration activity [85] compared to the static cultures. A decreased proliferation was also observed in bovine aortic SMCs seeded on glass slides collagen I-coated and exposed to laminar shear stress (11 dynes/cm^2^) for 24 h [86]. Unfortunately, there was no information about the SMC marker genes before and after the shear stress in all these studies.

Overall, the physiological relevance of the in vitro responses of SMCs to laminar shear stress is unclear. The patterns of shear stress at sites of endothelial cell injury in vivo do not necessarily mirror the continuous laminar shear stress addressed by several studies. Disturbed or turbulent shear stress patterns have been shown to induce atherosclerotic plaque formation in vivo and activate inflammatory signaling on endothelial cells in vitro [87]. However, the in vitro effects of disturbed or turbulent shear stress on the SMC phenotype have not been well-characterized. Pioneer studies have shown that bovine aortic SMC increased their DNA synthesis and proliferation capacity when exposed to oscillatory shear stress (14 dynes/cm^2^) for 3 or 5 days compared to the static controls [88], but the degree to which this was accompanied by changes in the SMC phenotypic markers was not analyzed. A more systematic characterization of the phenotype and function of SMCs exposed to a greater range of shear stress forces and patterns on relevant substrates are further required, Table 5.

## 6. Smooth Muscle Cell Mechanotransduction

The cellular process of converting mechanical cues into biochemical signals is known as cellular mechanotransduction. This aspect has been reviewed extensively in other vascular cells [89]. However, the precise mechanisms of cellular mechanotransduction on SMCs upon stretching are still not completely clear. In general terms, external mechanical forces can be transmitted to a cell in different ways, primarily by activating the integrin signaling pathway but also by G protein-coupled receptors (GPCRs), by nonselective cation channels, or by the coordinated and synergistic interactions of some or all of them [90]. The cytoplasmatic domain of integrins is functionally linked to various intracellular proteins such as talin, focal adhesion kinase (FAK), zyxin, paxillin, and vinculin. These proteins are organized as a focal adhesion complex to regulate the biochemical cascades initiated by mechanical forces [89]. The exposure of SMCs to physiological stretching (10%) for more than 6 h has shown increased levels of both the αv and β3 integrin subunits [46]. Experiments on rat SMCs have shown that stretching can induce cell adhesion kinase β, a highly related protein to FAK. Interestingly, this response seems to be partly mediated by the sodium and calcium ion channels [91]. At higher magnitudes of stretching (13% for 1 h), focal adhesion proteins such as zyxin are activated and translocated to the nucleus [92]. Other evidence suggests that integrins activate cellular responses upon stretching in coordination with the growth factor receptors [93]. The exposure of physiological cyclic (10%, 24 h) stretching on rat SMCs can also inhibit the Notch 3 receptor expression [94]. GPCRs have also been proposed to function as mechanoreceptors in SMCs. In particular, the angiotensin II type I receptor (AT_1_R) can be activated by excessive mechanical stretching (20%) and induce ERK signaling, which leads to increased migration and protein synthesis [54,95]. Different studies have shown that stretching can activate nonselective cation channels in SMCs. The stretch-induced increase of cytosolic calcium concentration in SMCs results from the release of intracellular calcium stores [96]. Cyclic stretching significantly decreases the transient receptor potential cation channel C4 (TRPC4) protein expression in SMCs [97]. Further studies are required to determine how these or potentially new mechanotransducers are regulated under normal and pathological stretching conditions on SMCs.

### Mechanotransduction Signaling Pathways in SMCs

Different studies have investigated the potential intracellular pathways induced by mechanical stretching on SMCs. For example, transforming growth factor-b (TGF-β) signaling has been implicated. The TGF-β1 levels were increased in the supernatant of rat SMCs exposed to 10% stretching for 24 h compared to the static controls [45]. Together with high TGF-β1 levels, the protein levels of ACTA2, CNN1, and TAGLN were also increased upon physiological cyclic stretching [45]. This study also showed that TGF-β1 could activate Smad2/5, leading to an increase of SIRT6 (a member of the sirtuin family). High levels of SIRT6 in the nuclei then mediate the upregulation of SM markers and, thus, a more contractile phenotype after stretching. Another member of the sirtuin family, SIRT1, was also shown to be upregulated during the physiological levels of stretching (10% for 24 h). SIRT1 promotes both the activation of forkhead transcription factor 3a (Foxo3a) and inhibition of Foxo4, resulting in a more contractile phenotype on stretched rat SMCs than the static controls [44]. The role of mechanical force-induced epigenetic modifications in vascular gene expression has been extensively studied in endothelial cells [98] and less in SMCs [99]. In rat smooth muscle cells, physiological stretching for 48 h (10%, 1 Hz) significantly regulated the expression of histone deacetylases, particularly HDAC3, 4, and 7, compared to static cultured cells (Figure 3A).

These changes in the expression of stretched cells accompanied a reduced migration compared to the static controls [100]. The other mechanisms by which shear stress and stretch induce the expression of epigenetic factors to modulate SMC functions have been recently reviewed [99].

Effectors of the Hippo pathway, YES-associated protein (YAP), and the transcriptional coactivator with a PDZ-binding motif (TAZ) are also involved in the stretch-induced phenotypic modulation of SMCs. YAP/TAZ activation after 24 h of cyclic stretching (13%) was linked to an increase in proliferation and proinflammatory gene expression (TNF-α, IL-6, IL-8, and IL-1B) in human umbilical artery SMCs compared to the static controls [49]. Human aortic SMCs subjected to stretching (16%) for 12 h increased their expression of angiotensin-converting enzyme (ACE), which, in turn, activated extracellular signal-regulated kinase-1 (ERK1). Phosphorylated ERK1 then blocked miR-145 and reduced the levels of the contractile marker genes inducing a phenotypic switch. In rat SMCs, the release of proinflammatory cytokine IL-6 was increased in cells subjected to 15% of cyclic stretching (from 3 to 24 h) compared to the static controls [56]. The authors described that this effect is mediated by a mechanism involving the Ras/Rac/p38 and NFKB signaling pathways (Figure 3B). Early works by others have also described the stimulation of RhoA by mechanical stress, but the mechanism is unknown [101,102].

These data exemplify the high degree of complexity among the intracellular pathways induced by stretching and highlighting the need for more research to identify new mechanoreceptors and regulators (Figure 3). Although several studies have tried to characterize the potential mechanoreceptors and intracellular pathways in SMCs, this aspect is still unclear.

## 7. Conclusions and Future Directions

Identifying the mechanisms that control SMC phenotypes is crucial to develop new drugs against vascular diseases characterized by the presence of highly modulated SMCs and for improvements in the tissue engineering of vascular tissues [103,104]. A better understanding of how mechanical forces and ECM proteins control the SMC phenotype is a particularly important part of that endeavour.

Despite clear evidence that mechanical forces can modulate SMC phenotypes, there are still many discrepancies in the results from different in vitro studies [16]. Some of those differences may be explained by variations in the stretching conditions such as the intensity, wave shape, duration, and frequency. Other factors that may contribute are culture conditions before and during stretching, including, in particular, ECM protein coating, as well as the variables pertaining to the cells under study, including the procedures for their isolation and propagation. There is a pressing need to define the conditions that recapitulate the in vivo conditions the best, and more complex in vitro culture systems, such as 3D cultures, may be needed to achieve that. A recent comparative study between the 2D and 3D models of the cyclic stretching stimulation of human SMCs revealed that a contractile protein expression was unchanged by stretching in 2D conditions but increased in a 3D collagen matrix [43]. The differences in the SMC responses observed in this study indicate that the dimensionality of the extracellular environment (2D or 3D) may be essential for the cellular responses to mechanical stimulation. Since many SMCs function in symbiosis with ECs, coculture systems of SMCs and ECs in the presence of mechanical stimuli and ECM substrates may also hold potential for new discoveries [105], as do models that can combine two or more types of mechanical forces in a single culture system [105,106]. To guide all of these optimizations, it will be necessary to conduct experiments that compare the in vitro findings with the effects of stretching on SMCs in vivo, e.g., achieved through surgical interventions in experimental animals.

Finally, the lack of thorough and unbiased approaches to investigate the effects of mechanical stimulation on SMC phenotypes limits the current research, which mostly relies on a few contractile markers. Recent studies using single-cell transcriptomics have demonstrated SMC heterogeneity in healthy and diseased arteries [107,108,109]. New transcriptomic analyses that integrate the in vitro models of mechanical forces are needed to unmask the potential mechanotransduction pathways beyond changes in the contractile machinery [110]. Furthermore, the determination of the phenotypic status of the SMCs must not rely exclusively on the assessment of transcriptomic changes. Other phenotypic switch features, such as cell morphology, proliferation, migration, and ECM production, do not necessarily depend on contractile gene expression and might be regulated by different mechanisms.

Considering the importance of stretch-induced changes in SMC phenotypes for physiological functions and disease processes and the many remaining knowledge gaps, clearly, more research in this area is needed.

## Figures and Tables

**Figure 1 cells-10-02209-f001:**
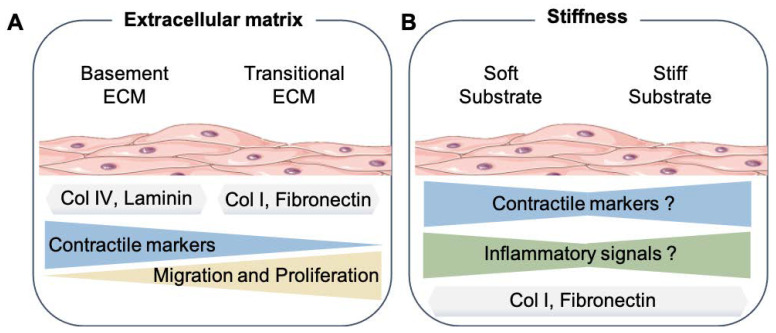
Smooth muscle cell mechanical microenvironment. (**A**) Summary of the results from different studies about the effects of basement extracellular matrix proteins (Col IV, Laminin, etc.) or transitional, also known as interstitial, extracellular matrix (Col I, Fibronectin, and Lumican) to the SMC phenotype. (**B**) How changes in the substrate stiffness can affect the SMC phenotype. Extracellular matrix proteins (ECM), Collagen type IV (Col IV), and Collagen type I (Col I). The SMC drawings were adapted from Servier Medical Art (SMART); https://smart.servier.com/ (accessed on 17 August 2021).

**Figure 2 cells-10-02209-f002:**
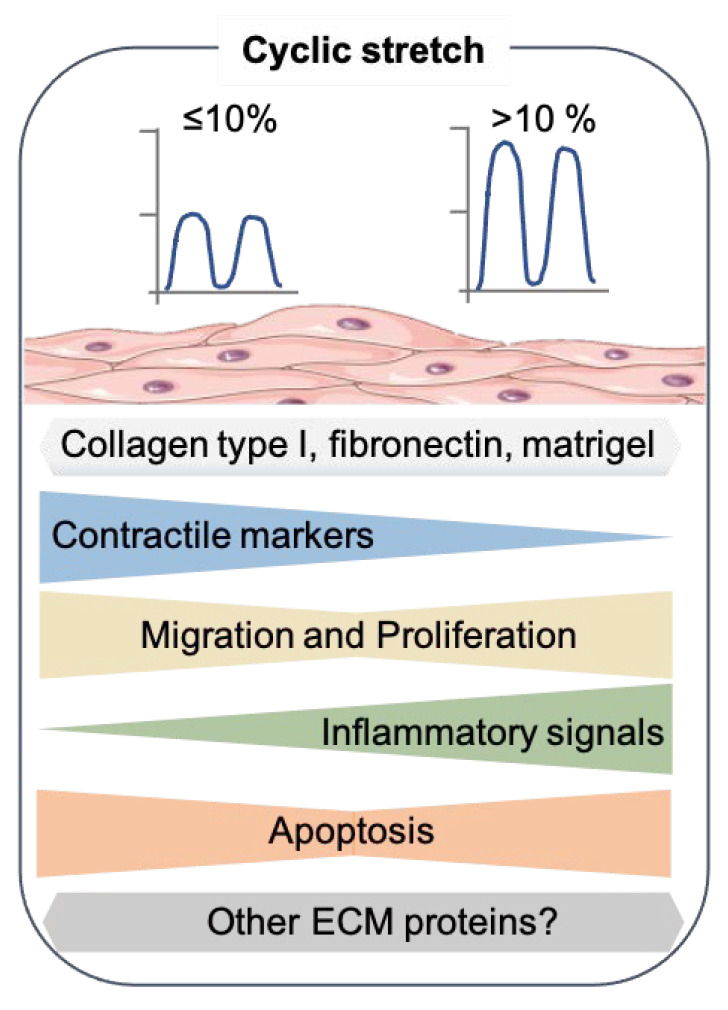
Effects of cyclic stretch on the SMC phenotype. Summary of the results from recent 2D in vitro studies that exposed the cyclic stretch to SMC. The SMC drawing was adapted from Servier Medical Art (SMART), https://smart.servier.com/ (accessed on 17 August 2021).

**Figure 3 cells-10-02209-f003:**
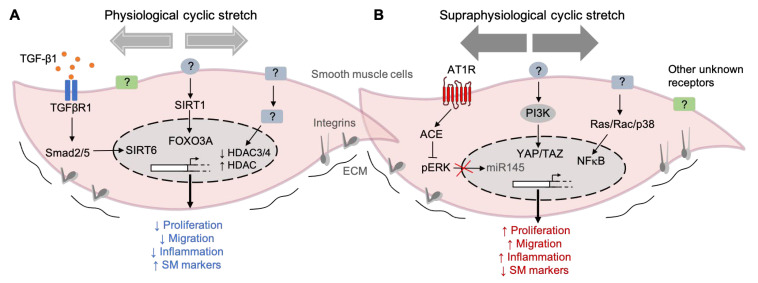
Examples of mechanical stretch-induced pathways in SMCs. (**A**) Pathways activated by physiological cyclic stretching (< 10% elongation) keep the SMCs into a differentiated state, characterized by a reduced proliferation, migration, and inflammation and accompanied by high levels of contractile marker genes. (**B**) On the other hand, the exposure of SMC to supraphysiological cyclic stretching (> 15% elongation) induces phenotypic modulation from a contractile to a synthetic state. The pathways activated by this high intensity stretching profile induce an increase in cell proliferation, migration, and inflammation and a decrease in the expression of contractile markers. (TGF-β1) Transforming growth factor-beta 1, (TGFBR1) Transforming growth factor-beta receptor 1, (Smad2/5) SMAD family members 2 and 5, (SIRT6) Sirtuin 6, (SIRT1) Sirtuin 1, (FOXO3) Forkhead transcription factor 3a, (HDAC) Histone deacetylase, (AT1R) Angiotensin II receptor type I, (ACE) angiotensin-converting enzyme, (pERK) phosphorylated extracellular-regulated protein kinase, (miR-145) microRNA 145, (PI3K) phosphoinositide 3-kinase, (YAP) Yes-associated protein 1, (TAZ) transcriptional coactivator with a PDZ-binding motif, (Ras/Rac) family of small GTPases, (P38) P38 mitogen-activated protein kinases, (NF-κB) nuclear factor kappa-light-chain-enhancer of activated B cells, (?) unknown receptor/regulator, and extracellular matrix (ECM).

**Table 1 cells-10-02209-t001:** Representative overview of recent in vitro 2D studies investigating the effect of cyclic stretch on the expression of SMC marker genes in human and rat SMCs. The Flexcell tension system was used in all the studies except for Reference [43], which used a custom-made device.

Study	Stretch Intensity, Duration and Frequency	Matrix Coating	SMC-Source	SM Marker Expression
[43]	7% for 2 and 5 days 1Hz	Collagen I	Human umbilical artery	*CNN1* (=) *ACTA2* (=)
[44]	10% for 24 h 1.25 Hz	Collagen I	Sprague–Dawley rat thoracic aorta	*CNN1* ↑ *ACTA2* ↑ *TAGLN* ↑
[45]	10% for 24 h 1.25 Hz	Gelatin	Sprague–Dawley rat thoracic aorta	*Cnn1* ↑ *Acta2* ↑ *Tagln* ↑
[46]	10% for 24 h 1 Hz	Collagen I	Wistar rat thoracic aorta	*Acta2* ↑ *Myh11* ↑
[47]	10% for 24 h 1 Hz	Collagen I	Sprague–Dawley rat thoracic aorta	*Cnn1* ↓ *Smtn* ↓ *Opn* ↑
[48]	13% for 24 h 0.5 Hz	Fibronectin	Human umbilical artery	*CNN1* ↓ *MYOCD* ↓ *MYH11* ↓
[49]	13% for 24 h 0.5 Hz	Matrigel	Human umbilical artery	*CNN1* ↓ *ACTA2* ↓ *MYH11* ↓
[50]	15% for 24 h 1.25 Hz	Collagen I	Sprague–Dawley rat thoracic aorta	*Cnn1* ↓ *Acta2* ↓ *Tagln* ↓
[51]	16% for 12 h 1 Hz	Collagen I	Human aorta	*CNN1* ↓ *ACTA2* ↓ *TAGLN* ↓ *MYOCD* ↓ *KLF4* ↑
[52]	20% for 24 h 1 Hz	Collagen I	Human aorta	*CNN1* ↓ *ACTA2* ↓

*ACTA2* (actin alpha 2 smooth muscle), *TAGLN* (transgelin), *OPN* (osteopontin), *CNN1* (Calponin), *KLF4* (Krüppel-like factor 4), and *MYOCD* (Myocardin). Unchanged (=), increased ↑, and decreased ↓.

**Table 2 cells-10-02209-t002:** Representative overview of recent in vitro 2D studies investigating the effect of cyclic stretch on human and rodent SMC migration. The Flexcell tension system was used in all these studies.

Study	Stretch Intensity, Duration and Frequeny	Matrix Coating	Technique Used	SMC- Source	Migration Effect
[58]	10% for 12 h 1 Hz	Collagen I	Scratch assay	Human aortic	Decreased
[47]	10% for 24 h 1 Hz	Collagen I	Scratch assay	Sprague–Dawley rat thoracic aorta	Increased
[59]	15% for 24 h 1.25 Hz	Collagen I	Transwell	Sprague–Dawley rat thoracic aorta	Increased
[60]	20% for 3 h, 1 Hz	Collagen I	Scratch assay	129/SV Mouse aortic	Increased

**Table 3 cells-10-02209-t003:** Representative overview of the recent in vitro 2D studies investigating the effect of cyclic stretching on human and rodent SMC proliferation. The Flexcell tension system was used in all these studies.

Study	Stretch Intensity, Duration and Frequeny	Matrix Coating	Technique Used	SMC Source	Proliferation Effect
[58]	10% for 12 h 1 Hz	Collagen I	BrdU incorporation	Human aorta	Decreased
[62]	10% for 48 h 1 Hz	Collagen I	Fluorescence spectroscopy	A7R5 rat thoracic aorta	Decreased
[61]	10% for 4 days, 1 Hz	Collagen I	Cell counts	Sprague–Dawley rat thoracic aorta	Decreased
[63]	10% for 1 h 1 Hz	Gelatin	Ki67 staining	C57BL/6J mouse aorta	Increased
[49]	13% for 24 h 0.5 Hz	Matrigel	EdU incorporation	Human umbilical artery	Increased
[59]	15% for 24 h 1.25 Hz	Collagen I	BrdU incorporation	Sprague–Dawley rat thoracic aorta	Increased
[66]	16% for 12 h 1 Hz	Collagen I	BrdU incorporation	Human aorta	Increased
[52]	20% for 24 h 1 Hz	Collagen I	Colorimetric assay	Human aorta	Increased

**Table 4 cells-10-02209-t004:** Representative overview of recent in vitro 2D studies that investigated the effect of cyclic stretching on human and rodent SMC apoptosis. The Flexcell tension system was used in all the studies except for References [74,75], which used STREX. Lactate dehydrogenase (LDH) and terminal deoxynucleotidyl transferase (TdT) dUTP nick-end labeling (TUNEL).

Study	Stretch Intensity, Duration, Frequency	Matrix Substrate	Technique Used	SMC Source	Apoptotic Effect
[72]	10% for 1–24 h 1 Hz	Gelatin	TUNEL	C57BL/6J Mouse aortic	Increased
[71]	10% for 1 h or 15 h 1 Hz	Gelatin	TUNEL	C57BL/6J Mouse aortic	Increased
[73]	10%for 1 h 1 Hz	Gelatin	TUNEL	C57BL/6J mouse aorta	Increased
[75]	15% for 4 h 1 Hz	Collagen I	LDHrelease	Sprague–Dawley rat thoracic aorta	Increased
[76]	15% for 4 h 1 Hz	Collagen I	apoptosis marker genes	Sprague–Dawley rat thoracic aorta	Increased
[66]	16% for 12 h 1 Hz	Collagen I	Cell sorting	Human aortic	Increased
[78]	18% for 36 h	Collagen I	Cell sorting	C57B/L6 Mouse aortic	Increased
[65]	18% for 12 h 1 Hz	Collagen I	Flow cytometry	Human aortic	Increased
[77]	20% for 18 h 1 Hz	Collagen I	Cell sorting TUNEL	Human coronary	Increased

**Table 5 cells-10-02209-t005:** Representative overview of the recent in vitro 2D studies that investigated the effect of shear stress on human, rat, and bovine SMC phenotypes. Increased ↑ and decreased ↓.

Study	Shear Stress Type, Intensity, and Duration	Material and Matrix Substrate	SMC Source	Technique Used	Effects on SM Phenotype
[80]	Laminar: 8 dynes/cm^2^ for 15 h	Plastic/ fibronectin	Sprague–Dawley rat thoracic aorta	Rotating disk	*Acta2* ↓ *Tagln* ↓ *Myh11* ↓ *Smtn* ↓ *Cnn1* ↓
[81]	Laminar: 12 dynes/cm^2^ for 24 h	Glass/ fibronectin	Human aorta	Parallel plate flow chamber	↑ proliferation ↓ inflammation
[82]	Laminar: 14 dynes/cm^2^ for 24 h	Not stated	Rat aortic	Parallel plate flow chamber	*Myh11* ↓ *Smtn* ↓ *Acta2* ↓
[83]	Laminar: 15 dynes/cm^2^ for 6, 12 and 24 h	Plastic/ coating not stated	Rat Brain arteries	Parallel plate flow chamber	↑ proliferation ↑ migration *Acta2* ↓ *Tagln* ↓
[84]	Laminar: 14 dynes/cm^2^ for 24 h	Plastic/ coating not stated	Sprague–Dawley Rat aortic	Parallel plate flow chamber	↓ proliferation
[85]	Laminar: 12 dynes/cm^2^ for 24 h	Glass/ coating not stated	Sprague–Dawley rat thoracic aorta	Parallel plate flow chamber	↓ proliferation ↓ migration
[86]	Laminar: 11 dynes/cm^2^ for 24 h	Glass/ Collagen I	Bovine aortic	Parallel plate flow chamber	↓ proliferation
[88]	Oscillatory: 14 dynes/cm^2^ for 3 and 5 days	Plastic/ Collagen I	Bovine aortic	Orbital shaker	↑ proliferation

## Data Availability

Not applicable.

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
