# Peer review of "The Phenotypic Responses of Vascular Smooth Muscle Cells Exposed to Mechanical Cues"

_cells, 2021, doi:10.3390/cells10092209_

Round 1
Reviewer 1 Report
In this review paper Jensen and colleagues discuss the effects of vascular environment (extracellular matrix composition and stiffness) and mechanical forces (cyclic stretch) on vascular smooth muscle cell (VSMC) phenotypic responses (gene expression, migration and proliferation). The paper provides a systematic review of published 2D in vitro studies that address these aspects of regulation of VSMC phenotype.
The paper highlights the challenges of modelling VSMC environment in vitro and discusses the need to optimise culture conditions to best mimic behaviour of VSMCs in vivo, by using 3D models, co-culture with endothelial cells and integration of multiple mechanical effects.
It is overall well written, timely and will provide a useful source of reference for researchers studying vascular responses to mechanical forces. However, it is quite narrowly focused; the effect of other mechanical forces sucha as shear stress is briefly discussed in terms of endothelial cells, but not in terms of direct of exposure of VSMCs to shear stress in the regions of endothelial desquamation, as well as interstitial flow-induced shear stress. There is no mention of:
1) VSMC apoptosis, which is also affected by stretch and closely associated with vascular remodelling and disease development;
2) Ex vivo models to study the effects of cyclic stretch;
3) Recent advances in understanding VSMC heterogeneity in healthy and diseased arteries by using single cell transcriptomics (for example Dobnikar et al, 2018; Wirka et al, 2019; Pan et al, 2020);
In addition, mechanotransduction pathways regulated by mechanical forces (cyclic stretch, in particular) are relatively superficially discussed and focusing on upstream players (integrins, GPCRs etc) rather than more downstream pathways and mechanisms, both transcriptional (including Yap/Taz pathway) and epigenetic mechanisms. An additional figure would be useful to summarise these different molecular pathways and VSMC responses.
Minor comments:
1) In Tables 1-3, column "Stretch intensity" should state "Stretch intensity, duration and frequency".
2) In section 4.2 a reference is missing after second sentence in paragraph 3 ("For instance, human aortic SMCs culture on...").
3) Do all studies listed in Tables 1-3 use Flexcell system to apply stretch to VSMCs? If not, it may be useful to include what system was used in each study as this may also account for some of the discrepancies observed between the studies.
4) Many of the references are not properly formatted (for example 3, 13-14, 16, 18-19 etc).
Author Response
Response to Reviewer 1 Comments
Reviewer 1 comments:
In this review paper Jensen and colleagues discuss the effects of vascular environment (extracellular matrix composition and stiffness) and mechanical forces (cyclic stretch) on vascular smooth muscle cell (VSMC) phenotypic responses (gene expression, migration and proliferation). The paper provides a systematic review of published 2D in vitro studies that address these aspects of regulation of VSMC phenotype. The paper highlights the challenges of modelling VSMC environment in vitro and discusses the need to optimise culture conditions to best mimic behaviour of VSMCs in vivo, by using 3D models, co-culture with endothelial cells and integration of multiple mechanical effects.
It is overall well written, timely and will provide a useful source of reference for researchers studying vascular responses to mechanical forces.
Point 1: However, it is quite narrowly focused; the effect of other mechanical forces sucha as shear stress is briefly discussed in terms of endothelial cells, but not in terms of direct of exposure of VSMCs to shear stress in the regions of endothelial desquamation, as well as interstitial flow-induced shear stress.
Response:
We appreciate the reviewer's comment. We have now included information about the effects of fluid shear stress on smooth muscle cell function (including a table summarizing representative findings) and the current and future challenges. The reviewer can find this new information on page 14, line 422
Point 2:
There is no mention of:
1) VSMC apoptosis, which is also affected by stretch and closely associated with vascular remodelling and disease development;
Response: We thank the reviewer for this observation. We have now included information about the effects of mechanical stretch and apoptosis. The reviewer can find this new information on page 11 line 379
2) Ex vivo models to study the effects of cyclic stretch;
Response: We thank the reviewer for this observation. For this review, we have decided to focus on in vitro models of mechanical forces, emphasizing smooth muscle cells. Ex vivo models are important and complex as in most of the cases have functional endothelial and smooth muscle layers (DOI:10.1016/j.carpath.2009.06.007). We believe ex vivo models represent a whole new topic that others can address in the future.
3) Recent advances in understanding VSMC heterogeneity in healthy and diseased arteries by using single cell transcriptomics (for example Dobnikar et al, 2018; Wirka et al, 2019; Pan et al, 2020);
Response: We thank the reviewer for this observation. We are well aware and excited about these critical studies in the SMC and atherosclerosis field. Unfortunately, these techniques have yet to be extended to studies of controlled stretch or shear stress and hence it is difficult to make the link between the topic of this review - in vitro models of mechanical forces – and scRNA-seq analysis of vascular SMC heterogeneity in vivo. We have added the following sentences to the manuscript (line 596):
“Recent studies using single cells transcriptomics have demonstrated SMC heterogeneity in healthy and diseased arteries (Dobnikar et al., 2018; Pan et al., 2020; Wirka et al., 2019). New transcriptomic analyses that integrate in vitro models of mechanical forces are needed to unmask potential mechanotransduction pathways beyond changes in the contractile machinery”
Point 3: In addition, mechanotransduction pathways regulated by mechanical forces (cyclic stretch, in particular) are relatively superficially discussed and focusing on upstream players (integrins, GPCRs etc) rather than more downstream pathways and mechanisms, both transcriptional (including Yap/Taz pathway) and epigenetic mechanisms. An additional figure would be useful to summarise these different molecular pathways and VSMC responses.
Response: Thank you. We have added new information about downstream pathways and mechanisms regulated by cyclic stretch. We have also included YAP/TAZ, and epigenetic mechanisms, as the reviewer pointed out. Following the reviewer's advice, this new information includes a new figure summarizing some of these intracellular signaling pathways. The reviewer can find this new information on line 513
Minor comments:
- In Tables 1-3, column "Stretch intensity" should state "Stretch intensity, duration and frequency".
Response: We thank the reviewer for this observation. All the tables have been changed accordingly.
- In section 4.2 a reference is missing after second sentence in paragraph 3 ("For instance, human aortic SMCs culture on...").
Response: We thank the reviewer for this observation. We have added the missing reference.
- Do all studies listed in Tables 1-3 use Flexcell system to apply stretch to VSMCs? If not, it may be useful to include what system was used in each study as this may also account for some of the discrepancies observed between the studies.
Response: Thank you. We have added information about the in vitro setup used in the studies on the table legends.
- Many of the references are not properly formatted (for example 3, 13-14, 16, 18-19 etc).
Response: We thank the reviewer for this observation. We have properly formatted all the references.
Reviewer 2 Report
Reviewer’s comments
Vascular smooth muscle cells (SMCs) undergo a phenotypic change from the differentiated contractile phenotype to the dedifferentiated synthetic proliferative phenotype during the development of atherosclerosis and other vascular diseases. This phenotypic response appears to be associated with reduced or lost proteins linked to contractility proteins, and upregulation of different proteins including matrix remodeling enzymes and expression of different types of extracellular matrix (ECM) proteins that support proliferative phenotype. Similar phenotypic changes occur when SMCs are removed from the vasculature and cultured in-vitro on the static plate surface, However, the influence of ECM environment and mechanical forces are not commonly defined.
The authors of the referenced article compiled and chronicled literature relating to implications of mechanical forces on vascular smooth muscle cell (SMC) phenotypic changes, and discussed the significance of mechanical forces and the resulting biochemical signals in changing vascular smooth muscle (SMC) phenotype.
Even though the studies reported in this article provides evidence that mechanical forces play a role in modulating SMC phenotype, there are many discrepancies between different in vitro studies warranting the need for optimization of several different conditions including culture condition, procedures used for the isolation of cells, type of ECM proteins, and intensity, duration, and frequency of the mechanical force to reproduce the in-vivo condition. These limitations are appropriately reported by the authors of the article.
Overall, the article is a well-written review focusing on the influence of mechanical forces on vascular SMCs cultures, in particular, the effects of mechanical forces on their interaction with ECM proteins, expression of contractile and synthetic phenotype-specific marker genes, and regulation of SMC functions including migration and proliferation. It also addresses the discrepancies observed in the studies included in the article.
Minor comments and suggestions:
- Define ‘SB’, an abbreviation at the first usage on the last paragraph of the ‘Introduction’ section.
- There are several references with no volume and page numbers. For e.g. #3, 13, 14, 15, 16….
- It would be nice if images of SMCs subjected to different cyclic stretches are included in the article.
Author Response
Response to Reviewer 2 Comments
Vascular smooth muscle cells (SMCs) undergo a phenotypic change from the differentiated contractile phenotype to the dedifferentiated synthetic proliferative phenotype during the development of atherosclerosis and other vascular diseases. This phenotypic response appears to be associated with reduced or lost proteins linked to contractility proteins, and upregulation of different proteins including matrix remodeling enzymes and expression of different types of extracellular matrix (ECM) proteins that support proliferative phenotype. Similar phenotypic changes occur when SMCs are removed from the vasculature and cultured in-vitro on the static plate surface, However, the influence of ECM environment and mechanical forces are not commonly defined.
The authors of the referenced article compiled and chronicled literature relating to implications of mechanical forces on vascular smooth muscle cell (SMC) phenotypic changes, and discussed the significance of mechanical forces and the resulting biochemical signals in changing vascular smooth muscle (SMC) phenotype.
Even though the studies reported in this article provides evidence that mechanical forces play a role in modulating SMC phenotype, there are many discrepancies between different in vitro studies warranting the need for optimization of several different conditions including culture condition, procedures used for the isolation of cells, type of ECM proteins, and intensity, duration, and frequency of the mechanical force to reproduce the in-vivo condition. These limitations are appropriately reported by the authors of the article.
Overall, the article is a well-written review focusing on the influence of mechanical forces on vascular SMCs cultures, in particular, the effects of mechanical forces on their interaction with ECM proteins, expression of contractile and synthetic phenotype-specific marker genes, and regulation of SMC functions including migration and proliferation. It also addresses the discrepancies observed in the studies included in the article.
Response: We appreciate the reviewer´s positive feedback on our review.
Point 1: Define ‘SB’, an abbreviation at the first usage on the last paragraph of the ‘Introduction’ section.
Response: We could not find the SB abbreviation that the reviewer mentioned. However, we found SM (smooth muscle). Is that perhaps what the reviewer means? We have described SM abbreviation at the first usage in the last paragraph of the introduction.
Point 2: There are several references with no volume and page numbers. For e.g. #3, 13, 14, 15, 16….
Response: We thank the reviewer for this observation. We have properly formatted all the references.
Point 3: It would be nice if images of SMCs subjected to different cyclic stretches are included in the article.
Response: For the moment, we do not have images of SMCs subjected to different cyclic stretches. But are working on that and perhaps soon we can share them in a publication.
Round 2
Reviewer 1 Report
The authors have addressed my comments and I am happy with the improvements.